# Association of reduced serum EGF and leptin levels with the pathophysiology of major depressive disorder: A case-control study

**Md. Sohan**[1], **M. M. A. Shalahuddin Qusar**[2], **Mohammad Shahriar**[1], **Sardar Mohammad Ashraful Islam**[1], **Mohiuddin Ahmed Bhuiyan**[1], **Md. Rabiul Islam**[1]*

**1** Department of Pharmacy, University of Asia Pacific, Farmgate, Dhaka, Bangladesh, **2** Department of Psychiatry, Bangabandhu Sheikh Mujib Medical University, Shahabagh, Dhaka, Bangladesh

* robi.ayaan@gmail.com

**Data Availability Statement:** All relevant data are within the paper.

## Abstract

### Background

Major depressive disorder (MDD) is a heterogeneous mental disorder having a very diverse course and causing a significant changes in daily life. Though the exact pathophysiology of depression is still not known, an alteration in the serum levels of cytokines and neurotrophic factors was seen in MDD subjects. In this study, we compared the serum levels of 'pro-inflammatory cytokine leptin and neurotrophic factor EGF' in healthy controls (HCs) and MDD patients. To make the findings more accurate, we eventually looked for a correlation between altered serum leptin and EGF levels and the severity of the disease condition.

### Methods

For this case-control study, about 205 MDD patients were enrolled from the Department of Psychiatry, Bangabandhu Sheikh Mujib Medical University, Dhaka, and about 195 HCs were enrolled from various parts of Dhaka. The DSM-5 was utilized to evaluate and diagnose the participants. The HAM-D 17 scale was used to measure the severity of depression. After collecting blood samples, they were centrifuged to produce clear serum samples. These serum samples were analyzed using enzyme-linked immunosorbent assay (ELISA) kits to measure serum leptin and EGF levels.

### Results

We observed lowered serum EGF levels in MDD patients compared to HCs (524.70 ± 27.25 pg/ml vs. 672.52 ± 49.64 pg/ml, $p = 0.009$), and HAM-D score was elevated in MDD patients compared to HCs (17.17 ± 0.56 vs. 2.49 ± 0.43, $p<0.001$). But no correlation was established between serum EGF levels and the severity of depression. However, no significant differences were observed between MDD patients and HCs in the case of serum leptin levels ($p = 0.231$).

### Conclusion

Our study findings suggest that reduced serum EGF levels have an impact on the pathogenesis of depression. But as per our investigation, the severity of depression is not correlated

**Funding:** The author(s) received no specific funding for this work.

**Competing interests:** The authors have declared that no competing interests exist.

with altered EGF levels. Our findings regarding the association of EGF with MDD would help to use EGF as a risk indicator of depression. We suggest further clinical investigations to determine the precise function of leptin and EGF in depression.

## Introduction

MDD is a heterogeneous disorder with a quite varied course and is distinguished by a low mood, lowered interest, and poor cognitive abilities [1, 2]. MDD is responsible for causing a drastic change in everyday life. At least a few psychological and physiological abnormalities, such as alteration in sleep, altered appetite, low sexual urges, the inability to enjoy activities at work, loss of enjoyment with friends, weeping, suicidal ideation, and a slowdown of speech, are all associated with MDD. The fundamental criteria for diagnosis of MDD are that such alterations must cause significant disruptions to work and family life for at least two weeks. In the development of MDD, a lot of factors can be related. According to an estimation, about 35% of MDD is genetically predisposed, which means that genetic factors may be involved in how the disease develops [1, 3–6]. Moreover, a variety of social factors, such as socioeconomic impression, lack of social support, along with poor educational background, considerably raise the risk of MDD in both males and females [7]. On the one hand, trauma regarding 'physical, emotional, or sexual abuse during childhood might also be a contributing factor to MDD [8]. In contrast, MDD is the second most significant medical condition when contributing to the burden of chronic diseases. Heart disease, diabetes mellitus, and stroke are all illnesses linked to an increased chance of developing [2, 9, 10]. It has been demonstrated in a study that more than 26.1% of females and 14.7% of males are dealing with depression during their lifespan in the USA [11]. This finding supports the conclusion made in a study that women are roughly twice as likely as males to experience MDD [12]. According to a survey report, the prevalence of MDD in a calendar year varied by country, which is about 2.20% and 10.40% in Japan and Brazil, respectively [13]. Another study has revealed that people from middle-income nations are less affected by MDD than people from high-income nations. Furthermore, it has also been found that the chance to affect by MDD is decreased with age in people from high-income countries [14]. In terms of Bangladesh, a study stated that the prevalence of depressive disorder was about 4.4% [15]. MDD is also linked to causing suicide. People with MDD are at risk of dying by suicide 20 times more than the general population [16, 17]. As MDD is one of the leading causes of the deterioration of personal and social life, thus knowledge about the progression of MDD is essential.

It has been challenging to understand the pathophysiology of MDD because of the clinical variability of the condition. Several hypotheses have been proposed till now about the development and progression of major depression. All of those hypotheses are supported by investigations regarding mental stress and findings about an increase in the level of hormones released due to stress like a CRH (corticotropin-releasing hormone) [1, 18], alteration in the levels of inhibitory and excitatory neurotransmitters such as GABA, glutamate [19], reduction in the availability of neurotransmitters, e.g., dopamine, noradrenaline, and serotonin in the synaptic cleft [1, 20], respectively. Also, it has been observed that the neurotrophic factors and circadian cycle have a role in the pathophysiology of depression [1, 21]. The neurotrophic factors have been found at a decreased level, usually in the case of MDD patients, along with animal models with MDD [22]. Thus, it is universally acknowledged that an imbalance in neurochemicals causes the pathophysiology of MDD. However, a change in neurochemical structure is also associated with major depression [23–25]. For the genesis of depression, the involvement of

inflammatory cytokines is also demonstrated in many studies. By influencing neural activity either directly or indirectly, inflammatory cytokines raise the likelihood of mood disorders, according to growing evidence. For instance, IFN-α a cytokine involved with the pathogenesis of depression by affecting the frontal lobe, dopaminergic, serotonergic, and glutamatergic activity [26–29]. According to the involvement of cytokines with the inflammatory process, they are classified as pro-inflammatory and anti-inflammatory [17]. Several studies have been conducted with pro and anti-inflammatory cytokines separately to understand the association of cytokines more precisely. It has been depicted that, an elevation in the levels of a few pro-inflammatory cytokines and a reduction in the levels of a few anti-inflammatory cytokines may be responsible for causing depression [30–33]. Thus, it can be concluded that alterations in the levels of cytokines and neurotrophic factors might be involved with the pathophysiology of depression.

Leptin and Epidermal growth factor (EGF) are neurochemicals that may have an impact on the development of MDD. The pro-inflammatory cytokine leptin, produced by adipocytes, plays a role in immune system functions. As a cytokine, it can influence thymic homeostasis and affect the secretion of TNF-α and IL-1, which are separately involved with the progression of depression [34–37]. Many clinical studies have looked into the connection between leptin and depression. However, results from all of these investigations indicate a discrepancy. Initially, studies have concluded after conducting case-control studies that leptin levels remain in low levels in depressive patients compared with healthy subjects [38–40]. In contrast, higher leptin levels were also found in MDD patients [41, 42]. Another case-control study showed that major depressive individuals exhibited higher leptin levels at night. The study hypothesized that elevated leptin levels at night increase CRH secretion, a hormone associated with the progression of depression [43]. In contrast, few case-control studies have not found any significant change in leptin levels in MDD patients. A meta-analysis also supports the inconsistent involvement of leptin with major depression [44–46]. In terms of EGF, it acts as a neurotrophic factor and has contributed to MDD development. A preliminary investigation has been conducted among MDD individuals to determine the levels of neurotrophic factors. It has been depicted that the plasma levels of EGF have decreased when compared to healthy controls [22]. In comparison, an increase in EGF levels was found in another case-control study [47]. Moreover, another study concluded that no significant difference was found after comparing plasma concentration of EGF levels between patients and control groups [48].

Thus, the involvement of leptin and EGF with MDD still needs to be clarified as changes in leptin and EGF plasma levels between patient and control groups exhibit discrepancies. As leptin acts as a proinflammatory cytokine, it can potentially induce the inflammatory process. It involves with depression in a different manner compared to EGF, as it acts as a neurotrophic factor. In this study, we aimed to investigate the association of leptin and EGF with MDD simultaneously. Thus, it will help to understand the pathophysiology of MDD from two different points of view. In order to draw a clear line between the involvement of leptin and EGF serum levels with MDD, we aimed to inspect leptin and EGF levels in MDD patients and healthy controls. In addition, if any variation would be discovered, we sought to look into the relationship between changed serum leptin and EGF levels and the severity of depression.

## Methods

### Study participants

We enrolled 205 individuals with MDD and 195 healthy controls (HCs) in this case-control study. Patients having MDD were enlisted from the Department of Psychiatry, Bangabandhu Sheikh Mujib Medical University, Dhaka, Bangladesh. From all around Dhaka, HCs were

recruited. Regarding sex and age, the HCs and MDD patients were almost identical. The DSM-5 was used by a professional psychiatrist to diagnose MDD patients. In order to evaluate the severity of depression, the HAM-D 17 scale was used. Information regarding socio-demographic variables and any other mental illness of MDD patients and HCs was collected by using a pre-structured questionnaire. In our investigation, we included participants ranging from 18 to 60 years old. Furthermore, those MDD patients were enrolled in whom depressive symptoms had been present for at least two weeks. We did not include participants who had ever experienced heart disease, hepatic failure, epilepsy, or renal failure. Participants undergoing antidepressant treatments that could affect the concentration of serum leptin and EGF levels were also omitted. Furthermore, those with further comorbid mental problems, chronic organic disorders, a very high BMI, substance use, or dependency were omitted from the study.

### HAM-D 17 scale

To measure the severity of major depressive disorder, the 17 item Hamilton Depression Rating Scale (HAM-D) was utilized. According to this scale, the total score is classified into four distinct categories: "no depression" when the score is between 0 to 7, "mild depression" when the score is between 8 to 16, "moderate depression" when the score is between 17 to 23, "severe depression" when the score is ≥24 [49].

### Blood sample collection

After collecting 10ml of blood from the cephalic vein of study subjects, blood was placed in centrifuge tubes. The serum was collected after centrifuging each sample at 1000 x g for 15 minutes and then stored at -80°C for future analysis.

### Analysis of serum samples

We utilized commercially available enzyme-linked immunosorbent assay (ELISA) kits to assess the serum levels of leptin and EGF based on the instructions made by the manufacturer (Boster Biological Technology, USA). The procedures are the same for leptin and EGF analysis. We used marker-specific reagents where necessary. Initially, the serum samples were brought to room temperature and vortexed to create a homogeneous mixture. Briefly, about 100μl of (diluted) standards and samples were placed in each well of microplates, where each well was coated with the capture antibody. After incubating microplates (sealed with plate sealer) at 37°C for one and half an hour, the liquid was removed to put 100μl of detection antibodies in wells. Again, the plates were incubated for one hour following the same incubation conditions, and after then, the liquids were discarded. Before putting 100μl of the avidin-biotin-peroxidase complex in each well, plates were washed three times using phosphate buffer solution. Then, the plates were incubated for 40 minutes at room temperature; the plates were rewashed using a wash buffer five consecutive times. After then, about 90μl of the color-developing agent was added to each well and then incubated each plate at room temperature in a dark area. In the end, about 100μl of stop solution was added, and within 30 minutes, absorbance was measured at 450nm using a microplate reader. The serum levels of leptin and EGF in MDD and HCs were calculated using a marker-specific standard curve and expressed in ng/ml and pg/ml, respectively.

### Statistical analysis

IBM SPSS (version 25.0) and Microsoft Excel 2019 were utilized to complete the necessary analysis. The Independent sample t-test and chi-square test were applied to differentiate

between the groups for variables (continuous and categorical). The box-plot graph 1 demonstrated the differences in the blood EGF concentrations, and box-plot graph 2 demonstrated the differences in the serum leptin concentrations of HCs and MDD patients. The distribution of EGF and leptin levels in MDD patients against the HAM-D score was depicted in scatterplot graph 1 and scatterplot graph 2, respectively. The socio-demographic factors of the study subjects were subjected to descriptive statistical calculations, and the findings were presented as mean ± standard error mean (SEM). The findings were considered statistically significant when the $p$-value was less than 0.05.

## Ethics

The Research Ethics Committee (REC) of the University of Asia Pacific approved the research protocol (Ref: UAP/REC/2022/206). All participants were informed regarding the study's purpose, and their written consent was acquired. We conducted each investigation in accordance with the Declaration of Helsinki.

## Results

### Characteristics of study population

The socio-demarchic and biographical characteristics of the study participants are demonstrated in Table 1. No significant difference was observed between MDD patients and HCs in terms of age ($p$ = 0.901) as the number of males and females was almost equal in both groups. Apart from that, significant differences were observed in case of marital status, educational level, occupation, economic impression, area of residence, previous history and family history of MDD.

### Clinical profile and laboratory findings

The clinical profile and laboratory findings are presented in Table 2. The DSM-5 score was 7.49 ± 0.087 among MDD patients, and the DSM-5 score was 1.32 ± 0.098 in HCs ($p$<0.001). The HAM-D score was 17.97 ± 0.358 and 2.25 ± 0.209 in MDD and healthy subjects, respectively ($p$<0.001). Lower leptin levels were observed in MDD patients (62.83 ± 4.28 ng/ml) in comparison with HCs (69.74 ± 3.85 ng/ml), $p$ = 0.231. The serum levels of EGF were 524.70 ± 27.25 pg/ml in MDD patients and 672.52 ± 49.64 pg/ml in HCs ($p$ = 0.009). Moreover, the boxplots exhibit the distribution of serum EGF and leptin levels in MDD patients and HCs (Fig 1). It shows higher EGF and leptin levels in HCs than in MDD patients. On the other hand, sex-specific scatter plots depict the distribution of serum EGF and leptin levels in males and females with MDD against the HAM-D score (Fig 2).

## Discussion

In this study, we investigated the involvement of EGF (a neurotrophic factor) and leptin (a pro-inflammatory cytokine) with MDD and depression severity. We measured the serum levels of leptin in MDD patients compared to HCs. The serum leptin levels were lower in MDD patients than HCs, but the alteration was not significant ($p$ = 0.231). Though many research has been conducted to determine the relationship between leptin and depression, a concrete association was not established yet. A few case-control studies have suggested that reduced serum leptin levels are observed in depressive patients [38–40]. However, few other studies suggested that leptin levels were higher in depressive patients than in control subjects [42, 50]. Clinical research revealed that elevated serum leptin levels in young people with major depression were linked with the degree of depression [51]. Comparatively, a few case-control

**Table 1. Socio-demographic profile of the study population.**

| Characteristics | MDD patients (n = 205) Mean ± SEM | Healthy controls (n = 195) Mean ± SEM | *p-value* |
|---|---|---|---|
| Age in years | 30.80 ± 0.671 | 30.08 ± 0.663 | 0.901 |
| 18–25 | 76 (37.07%) | 67 (34.36%) | |
| 26–35 | 75 (36.59%) | 73 (37.44%) | |
| 36–45 | 36 (17.56%) | 39 (20.00%) | |
| 46–60 | 18 (8.78%) | 16 (8.21%) | |
| Sex | | | 0.754 |
| Male | 65 (31.71%) | 59 (30.26%) | |
| Female | 140 (68.29%) | 136 (69.74%) | |
| Marital Status | | | 0.001 |
| Married | 134 (65.37%) | 96 (49.23%) | |
| Unmarried | 71 (34.63) | 99 (50.77%) | |
| BMI (kg/m$^2$) | 23.57 ± 0.308 | 24.45 ± 0.261 | 0.006 |
| Below 18.5 (CED) | 21 (10.24%) | 5 (2.56%) | |
| 18.5–25 (normal) | 113 (55.12%) | 109 (55.90) | |
| Above 25 (obese) | 71 (34.63%) | 81 (41.54%) | |
| Education level | | | 0.020 |
| Illiterate | 16 (7.80%) | 15 (7.69%) | |
| Primary level | 50 (24.39%) | 29 (14.87%) | |
| Secondary level | 77 (37.56%) | 66 (43.59%) | |
| Graduate and above | 62 (30.24%) | 85 (56.32%) | |
| Occupation | | | < 0.001 |
| Business | 7 (3.41%) | 4 (2.05%) | |
| Service | 34 (16.59%) | 57 (29.23%) | |
| Housewife | 46 (22.44%) | 23 (11.79%) | |
| Unemployed | 56 (27.32%) | 36 (18.46%) | |
| Student | 27 (13.17%) | 59 (30.26%) | |
| Others | 35 (17.07%) | 16 (8.21%) | |
| Economic impression | | | < 0.001 |
| High | 24 (11.71%) | 81 (41.54%) | |
| Medium | 148 (72.20%) | 108 (55.38%) | |
| Low | 33 (16.10%) | 6 (3.08%) | |
| Smoking habit | | | 0.875 |
| Non-smoker | 187 (91.22%) | 177 (90.77%) | |
| Smoker | 18 (8.77%) | 18 (9.23%) | |
| Residence area | | | < 0.001 |
| Rural | 87 (42.44%) | 45 (23.08%) | |
| Urban | 118 (57.56%) | 150 (76.92%) | |
| Previous history of MDD | | | < 0.001 |
| Yes | 111 (54.15%) | 0 (0.00%) | |
| No | 94 (45.85%) | 195 (100.00%) | |
| Family history of MDD | | | < 0.001 |
| Yes | 58 (28.29%) | 2 (1.03%) | |
| No | 147 (71.71%) | 193 (98.97%) | |

Abbreviations: BMI, body mass index; CED, chronic energy deficiency; MDD, major depressive disorder; SEM, standard error mean. (*p*-value was determined by chi-square test)

**Table 2. Clinical profile and laboratory findings of the study population.**

| Parameters | MDD patients (n = 205)<br>Mean ± SEM | Healthy controls (n = 195)<br>Mean ± SEM | *p-value* |
|---|---|---|---|
| Age (years) | 30.80 ± 0.671 | 30.96 ± 0.663 | 0.901 |
| Male (P/C:65/59) | 29.95 ± 1.23 | 31.41 ± 1.24 | 0.407 |
| Female (P/C:140/136) | 31.19 ± 0.81 | 30.76 ± 0.79 | 0.703 |
| BMI (Kg/m$^2$) | 23.57 ± 0.308 | 24.45 ± 0.261 | 0.006 |
| Male (P/C:65/59) | 23.42 ± 0.55 | 24.55 ± 0.45 | 0.118 |
| Female (P/C:140/136) | 23.64 ± 0.37 | 24.41 ± 0.32 | 0.119 |
| DSM-5 score | 7.49 ± 0.087 | 1.32 ± 0.098 | < 0.001 |
| Male (P/C:65/59) | 7.60 ± 0.16 | 1.37 ± 0.18 | < 0.001 |
| Female (P/C:140/136) | 7.44 ± 0.11 | 1.29 ± 0.12 | < 0.001 |
| Ham-D score | 17.97 ± 0.358 | 2.25 ± 0.209 | < 0.001 |
| Male (P/C:65/59) | 17.17 ± 0.56 | 2.49 ± 0.43 | < 0.001 |
| Female (P/C:140/136) | 18.34 ± 0.45 | 2.15 ± 0.23 | < 0.001 |
| Serum leptin (ng/ml) | 62.83 ± 4.28 | 69.74 ± 3.85 | 0.231 |
| Male (P/C:65/59) | 74.38 ± 9.67 | 73.52 ± 8.07 | 0.947 |
| Female (P/C:140/136) | 57.49 ± 4.32 | 68.10 ± 4.28 | 0.082 |
| Serum EGF (pg/ml) | 524.70 ± 27.25 | 672.52 ± 49.64 | 0.009 |
| Male (P/C:65/59) | 588.39 ± 56.07 | 519.65 ± 56.29 | 0.390 |
| Female (P/C:140/136) | 495.13 ± 30.06 | 738.84 ± 66.18 | 0.001 |

Abbreviations: BMI, body mass index, DSM-5, diagnostic and statistical manual for mental disorders, 5[th] edition; Ham-D, 17-item Hamilton depression rating scale; EGF, epithelial growth factor; MDD, major depressive disorder; P/C, patients/control; SEM, standard error mean. (*p*-value was determined by independent samples t-test)

investigations on MDD patients failed to detect any appreciable alteration in leptin levels. The ambiguous relationship between serum leptin levels and depression was also supported by a meta-analysis of depression [44–46]. The fact that some variables, including sex, age, body mass status, sample sizes, and co-morbidity with other illnesses, impact serum leptin levels and these variables could be one of the reasons for the discrepant results. Another reason could be that leptin alteration may only be present in a subset of people who are depressive [52].

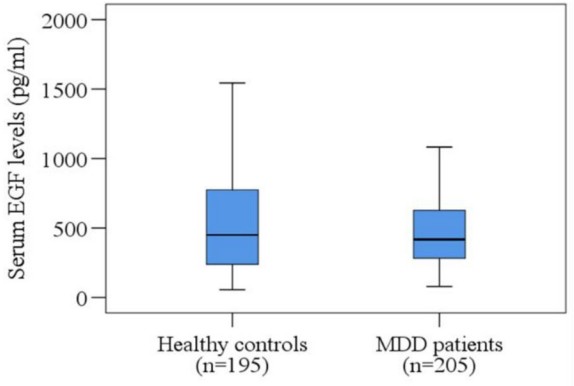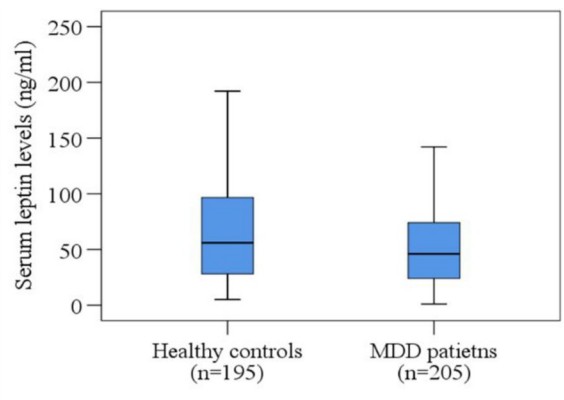

**Fig 1. Distribution of serum EGF and leptin levels in MDD patients and healthy controls.** Boxplot graphs showing the median, maximum and minimum value range.

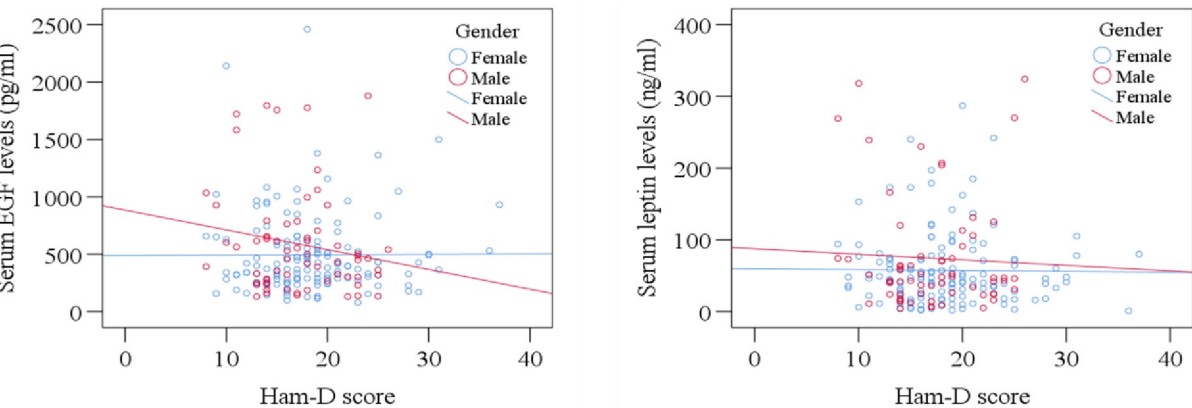

**Fig 2. Sex-specific scatter plot graphs showing distribution of EGF and leptin levels against HAM-D scores of MDD patients.**

On the other hand, in this study, we also measured the serum levels of EGF in MDD patients compared to HCs. We found a significant decrease in EGF levels among MDD patients ($p = 0.009$). A substantial increase in DSM-5 score was also observed in MDD patients, as expected ($p < 0.001$). Similarly, the HAM-D score was also higher in MDD patients than HCs, and the value was statistically significant ($p < 0.001$). So, it could be assumed that there might be a relation between the HAM-D score and serum EGF concentration as their values increased and decreased, respectively, in MDD patients against healthy populations. But the Pearson correlation study showed that the serum EGF levels and the severity of depression were not inversely proportional ($r = -0.064$, $p = 0.364$).

On the other hand, the association of EGF levels with depression was shown a very few studies. The levels of EGF in people with MDD have been the subject of a case-control preliminary inquiry. The serum EGF levels have been shown to be lower in MDD patients than in HCs [22]. Another study that compared serum concentrations of EGF levels between depressive patients and healthy subjects concluded that there was no noticeable change [48]. In contrast, another case-control investigation discovered increased EGF levels [47].

Furthermore, from sex-specific scatter plot graph where the distribution of serum EGF levels (pg/ml) against HAM-D score was depicted, it could be observed that the EGF levels were decreased in females more with the depression severity compared to males. After laboratory analysis, it was observed that the HAM-D score was higher in females than in males. At the same time, it was proven that serum EGF levels were lower in MDD females when compared to MDD males. But there was no significant correlation between sex and EGF levels.

Moreover, to the best of our knowledge, this investigation is the first attempt in Bangladesh to find out the involvement of serum leptin and EGF levels with depression and to find out their association with the severity of depression. Under the same experimental conditions, simultaneous serum leptin and EGF levels were measured in MDD patients and HCs. In addition, the levels of leptin and EGF in the blood of MDD patients were examined according to sex. The results of this investigation will help to determine the depression risk in the future. At the same time, this study will add extra value to the knowledge resources in understanding the pathophysiology of major depression. Despite several notable findings, our study has certain limitations. The serum levels of leptin and EGF were measured once after enrolling them. Measuring the serum leptin and EGF levels alone cannot accurately represent the entire neuroinflammatory process of major depressive disorder. It would help to understand the pathophysiology of MDD more accurately when measuring other parameters in the same demographic and laboratory settings [53, 54]. Moreover, the duration of depression, that

could potentially influence the levels of inflammatory parameters was not considered. The dietary habits of study populations, along with sleeping patterns, use of current medicines, or tobacco use, should have been taken into serious consideration in this present study, which might have an effect in altering serum leptin and EGF levels. Further investigation is suggested using larger samples, repeated measurements, and addressing this study's limitations.

## Conclusion

In this investigation, it was shown that there was a relationship between serum EGF levels and major depression. But involvement of the HAM-D score with serum EGF levels was not found in our investigation, thus a bridge between the severity of depression and EGF concentration could not be established. On the other hand, the involvement of leptin was not found in this study. Thus, after considering this study's findings as a possible candidate for depression and a risk indicator of MDD, we would like to suggest EGF. The results of this study might be helpful in managing MDD in the future. It is advised that larger sample sizes be used in future studies.

## Acknowledgments

Authors are thankful to participants of this study. They would like to thank the administrative staff and physicians of the department of psychiatry, Bangabandhu Sheikh Mujib Medical University, Dhaka, Bangladesh, for their cooperation to this study.

## Author Contributions

**Conceptualization:** Md. Sohan, Md. Rabiul Islam.

**Data curation:** Md. Sohan, M. M. A. Shalahuddin Qusar.

**Formal analysis:** Md. Sohan, Md. Rabiul Islam.

**Investigation:** Mohammad Shahriar, Sardar Mohammad Ashraful Islam, Mohiuddin Ahmed Bhuiyan.

**Methodology:** Md. Sohan, Md. Rabiul Islam.

**Project administration:** Mohammad Shahriar, Sardar Mohammad Ashraful Islam, Mohiuddin Ahmed Bhuiyan.

**Supervision:** Md. Rabiul Islam.

**Writing – original draft:** Md. Sohan.

**Writing – review & editing:** M. M. A. Shalahuddin Qusar, Md. Rabiul Islam.

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
