## [Decision Letter · Decision Letter 0]

24 Apr 2023

PONE-D-23-05701Reduced serum EGF but not leptin levels are associated with the pathophysiology of major depressive disorder: A case-control studyPLOS ONE

Dear Dr. Islam,

Thank you for submitting your manuscript to PLOS ONE. After careful consideration, we feel that it has merit but does not fully meet PLOS ONE’s publication criteria as it currently stands. Therefore, we invite you to submit a revised version of the manuscript that addresses the points raised during the review process.

ACADEMIC EDITOR:I have reviewed all the comments from reviewers and i think this paper requires "extensive modifications". I would invite the authors to submit a revised paper addressing all the comments from reviewers. 

We look forward to receiving your revised manuscript.

Kind regards,

Md Maruf Ahmed Molla

Academic Editor

PLOS ONE

Journal Requirements:

Reviewers' comments:

Reviewer's Responses to Questions

**Comments to the Author**

1. Is the manuscript technically sound, and do the data support the conclusions?

Reviewer #1: Partly

Reviewer #2: Yes

Reviewer #3: Yes

Reviewer #4: Partly

2. Has the statistical analysis been performed appropriately and rigorously? 

Reviewer #1: Yes

Reviewer #2: I Don't Know

Reviewer #3: Yes

Reviewer #4: No

3. Have the authors made all data underlying the findings in their manuscript fully available?

Reviewer #1: No

Reviewer #2: Yes

Reviewer #3: Yes

Reviewer #4: No

4. Is the manuscript presented in an intelligible fashion and written in standard English?

Reviewer #1: No

Reviewer #2: Yes

Reviewer #3: Yes

Reviewer #4: No

5. Review Comments to the Author

Reviewer #1: The manuscript is developed with an interesting subject

Lots of Acronyms used without elaboration

In title the acronym should be avoided

The abstract start with MDD is not understanding for many of the readers

In abstract the objective is not clearly mentioned

The result section is very small, need to be elaborated

The Odds Ratio (OR) is not analysed in the manuscript, but for case control study it is essential

Reviewer #2: Dear authors,

Congratulations on your hard work. However, I have some observations that may help with your manuscript.

1. Title: you have stated the result in the title. I might suggest to keep it minimum, like- Association of Reduced serum EGF and leptin levels with the pathophysiology of major depressive disorder: A case-control study

2. Abstract, background, line 3: please rephrase the line: this study, we compared the serum levels of ' pro-inflammatory cytokine leptin and neurotrophic factor EGF' of healthy controls (HCs) and

MDD patients.

3. Abstract, and background: please avoid abbrivations in the abstract. Also state the full term of MDD in the first line of background..

4. Rephrase the line: On the one hand, trauma regarding

'physical, emotional, or sexual abuse' during childhood might also be a contributing factor to MDD

5. Please add some prevalence of MDD in Bangladeshi background also

6. In background, add the rationale of exploring EGF and leptin level in MDD in practical purpouse. Will they help in the treatment of MDD?

7. Methods: State the hypothesis if you have any

8. Methods: add brief description about HAM-D scale. How it is calculated/scored

9. Discussion: state the possible practical application of your finding at the end of discussion

10. Table 1 & 2: state the statistical tests used to determine the p value under the tables

Reviewer #3: Thank you for inviting me to review the research paper titled “Reduced serum EGF but not leptin levels are associated with the pathophysiology of major depressive disorder: A case-control study “

I am writing to share my accepting of this study on the pathophysiology of depression. The research sheds important light on the complex mechanisms underlying depression and makes a significant contribution to the field. Even though the findings are still controversial and inconclusive yet but I commend the authors on the scientific rigor and meticulousness with which they presented the discussion, conclusions, recommendations, and limitations. Their attention to detail and thoughtful analysis demonstrate a deep understanding of the complexities of depression and reflect noted expertise and commitment to advancing the field. These findings have the potential to help future research and improve our understanding of depression, ultimately leading to better treatments and outcomes for those affected.

Well done,

Reviewer #4: The current manuscript aims to investigate whether the EGF and Leptin serum levels differ in MDD group from HCs. While I think that the research idea is very interesting, some changes seem to be essential for improving the academic quality of manuscript. Please see my comments below. Anyway, be it for resubmission in this journal or a submission to a different journal, I think the manuscript can be strengthened on condition of major revisions concerning all parts of the manuscript. I wish the authors the best of luck with this and their future research.

Introduction

1- The writing style of Reference 1,2,3. etc.. seem not to meet to reference rules..( All references need to be reviewed.

2- Many of the cited articles are rather old and there is newer evidence that should be considered and given the credit it deserves, for instance 10, 21-24, 25, 27-29, 31..( I strongly recommend that the references be reviewed for currency)

3- “According to a survey report, the prevalence of MDD in a calendar year varied by country, which is about 2.20% and 10.40% in Japan and Brazil, respectively. Another study has revealed that people from middle-income nations are more affected by MDD than people from high-income nations.”

“All of those hypotheses are supported by investigations regarding mental stress and findings about an increase in the level of hormones released due to stress like a CRH (corticotropin-releasing hormone), reduction in the availability of neurotransmitters, e.g., dopamine, noradrenaline, and serotonin in the synaptic cleft, alteration in the levels of inhibitory and excitatory neurotransmitters such as GABA, glutamate, respectively.”

can you provide a reference supporting these claims?

4- I think that the authors did not provide enough rationale for why they focused on EGF and Leptin together. As far as it is understood from the last 2 paragraphs, they preferred to emphasize only discrepancy, but the presence of contradictions in the relationship between pro- and anti-inflammatory parameters and depression in general seems to make this background somewhat inadequate. Indeed, to the extent that it is essential to consider the two parameters together, the authors should propose a solution to the problem of multiple comparisons (type-II error) in the analyses in which they are included as dependent variables. The section on the epidemiology of depression in the first half seems to overshadow the intended message to some extent. Instead, I would have preferred to expand the framework on why Leptin and EGF were included together, and a separate paragraph on the hypotheses would have made it easier for readers. If the authors take this route, I believe that whether they present their hypotheses on Leptin and EGF as the same or separately will influence the results of the study.

Method

1- The authors should indicate which treatments they excluded with the statement "Patients undergoing any therapy that could affect the concentration of serum leptin and EGF levels". For example, if they included the cases under antidepressant treatments, is the association with these two parameters at a level that requires to be considered as a control factor? Adding the basics of these to the "Introduction" would have complemented the biological framework of the article. Results

1- Table 2 shows that among male participants, serum EGF levels did not differ statistically from HC in MDD patients. The authors seem to have no indication of a gender effect, which I recommend controlling for in a model.

2- The information displayed in Table 1 seems to be (at least) redundant to the text in the Results part. Please decide whether you want to report it either in the main text, or in the table, but not both.

3- Adding correlation slopes in the scatter plot would make the visualization more meaningful. Also, I don't think the scatter plot is the appropriate graph to understand the “more severe depression levels of female MDDs”. Presenting the correlation coefficients in the text or in the graph will achieve the main goal here.

4- In addition, the characteristics of the study population are still given more space than the main results, and if they are to be presented in the text, it is more appropriate to summarize them. Instead, I suggest adding the points I mentioned to the main results.

Discussion

I assume that there might be a few changes to the discussion part based on my comments on the other parts, so I refrain from repeating my previously stated thoughts for all other parts of the manuscript and just focus on some additional comments I had for preparing a revision.

1- In the first paragraph, the authors begin the discussion section with a theoretical background that they have already included in the introduction. This seems to cause some disconnection in the narrative flow. Instead, I would suggest that the second paragraph should be placed first and the framework in the first paragraph should still be used in other parts of the discussion if necessary. I also think it is important that they state the extent to which the hypotheses I suggest to be presented in the introduction have been confirmed.

2- The statements in paragraph 3 “On the one hand, in this study, we also measured the serum levels of EGF in MDD patients compared to HCs. We found a significant decrease in EGF levels among MDD patients (p=0.009). A substantial increase in DSM-5 score was also observed in MDD patients, as expected (p<0.001) …...” do not infer a correlation between HAM-D and EGF concentration. It is essential to conduct the correlation analysis itself, rather than interpreting the results that allegedly lead to the inference of correlation.

3- From the statements in paragraph 4 “Furthermore, from sex-specific scatter plot graph where the distribution of serum EGF levels (pg/ml) against Ham-D score was depicted, it could be observed that the severity of depression was higher in females compared to males. After laboratory analysis, it was observed that the Ham-D score was higher in females than males, and the score difference between female subjects was higher than the score difference between male subjects. At the same time, serum EGF levels were lower in MDD females when compared to MDD males.”, it is clear that the authors have only made an observational interpretation. The statement that EGF causes depression more in women sounds ambitious on several counts. First, since the study design is cross-sectional, causality between variables cannot be established. The other is that the EGF concentration-severity of depression-gender relationship suggested by the authors can only be confirmed/falsified within an interaction model (e.g. factorial analysis of variance).

4- Limitation: I would expact that the authors add to the limitations that the duration of depression that could potentially influence the levels of inflammatory parameters.

6. PLOS authors have the option to publish the peer review history of their article (what does this mean?). If published, this will include your full peer review and any attached files.

Reviewer #1: No

Reviewer #2: No

Reviewer #3: No

Reviewer #4: No

---

## [Editor Report · Decision Letter 1]

20 Jun 2023

Association of reduced serum EGF and leptin levels with the pathophysiology of major depressive disorder: A case-control study

PONE-D-23-05701R1

Dear Dr. Islam,

We’re pleased to inform you that your manuscript has been judged scientifically suitable for publication and will be formally accepted for publication once it meets all outstanding technical requirements.

Kind regards,

Md Maruf Ahmed Molla

Academic Editor

PLOS ONE
---

## [Editor Report · Acceptance letter]

26 Jun 2023

PONE-D-23-05701R1 

Association of reduced serum EGF and leptin levels with the pathophysiology of major depressive disorder: A case-control study 

Dear Dr. Islam:

I'm pleased to inform you that your manuscript has been deemed suitable for publication in PLOS ONE. Congratulations! Your manuscript is now with our production department. 

Kind regards, 

on behalf of

Dr. Md Maruf Ahmed Molla 

Academic Editor

PLOS ONE